# On the Use of a Single Beam Acoustic Current Profiler for Multi-Point Velocity Measurement in a Wave and Current Basin

**DOI:** 10.3390/s20143881

**Published:** 2020-07-12

**Authors:** Marilou Jourdain de Thieulloy, Mairi Dorward, Chris Old, Roman Gabl, Thomas Davey, David M. Ingram, Brian G. Sellar

**Affiliations:** 1School of Engineering, Institute for Energy Systems, The University of Edinburgh, Max Born Crescent, Edinburgh EH9 3BF, UK; m.dorward@ed.ac.uk (M.D.); C.Old@ed.ac.uk (C.O.); brian.sellar@ed.ac.uk (B.G.S.); 2School of Engineering, Institute for Energy Systems, FloWave Ocean Energy Research Facility, The University of Edinburgh, Max Born Crescent, Edinburgh EH9 3BF, UK; roman.gabl@ed.ac.uk (R.G.); tom.davey@flowave.ed.ac.uk (T.D.); david.ingram@ed.ac.uk (D.M.I.)

**Keywords:** offshore renewable energy, tank testing, Acoustic Doppler Profiling, tidal current

## Abstract

Harnessing the energy of tidal currents has huge potential as a source of clean renewable energy. To do so in a reliable and cost effective way, it is critical to understand the interaction between tidal turbines, waves, and turbulent currents in the ocean. Scaled testing in a tank test provides a controlled, realistic, and highly reproducible down-scaled open ocean environment, and it is a key step in gaining this understanding. Knowledge of the hydrodynamic conditions during tests is critical and measurements at multiple locations are required to accurately characterise spatially varying flow in test tank facilities. The paper presents a laboratory technique using an acoustic velocimetry instrument, the range over-which measurements are acquired being more akin to open water applications. This enables almost simultaneous multi-point measurements of uni-directional velocity along a horizontal profile. Velocity measurements have been obtained from a horizontally mounted Single Beam Acoustic Doppler (SB-ADP) profiler deployed in the FloWave Ocean Energy Research Facility at the University of Edinburgh. These measurements have been statistically compared with point measurements obtained while using a co-located Acoustic Doppler Velocimeter (ADV). Measurements were made with both instruments under flow velocities varying from 0.6 ms^−1^ to 1.2 ms^−1^, showing that flow higher than 1 ms^−1^ was more suitable. Using a SB-ADP has shown the advantage of gaining 54 simultaneous measurement points of uni-directional velocity, covering a significant area with a total distance of 10 m of the test-tank, at a measurement frequency of 16 Hz. Of those measurement points, 41 were compared with co-located ADV measurements covering 8 m of the profile for a tank nominal flow velocity of 0.8 ms^−1^, and four distributed locations were chosen to to carry out the study at 0.6 ms^−1^, 1.0 ms^−1^, and 1.2 ms^−1^. The comparison with the ADV measurement showed a 2% relative bias on average.

## 1. Introduction

The exploitation of marine energy to support the transition to a low carbon future requires an extensive understanding of the harsh and complex ocean environment. Such an understanding facilitates the efficient design, cost-effective install, and reliable operation of tidal turbines. Test tanks that can recreate waves and currents at scale (representing the open ocean environment in a controlled, realistic, and highly reproducible manner) enables turbine testing to be performed at small scales (~1/30–1/15). This supports companies and their investors without the need to make the significant financial commitments needed to construct and deploy full-scale devices [1,2].

The experimental knowledge of the effect of flow on turbine loads and generated power enables the validation of numerical models that could simulate more complex cases than those that could be experimentally tested. Maganga et al. [3] presented measurement of the thrust and generated power from a single turbine, which were synchronised with flow measurements for various inflow conditions from 0.5 ms^−1^ to 1.5 ms^−1^ with two ambient turbulent intensities (TI) and compared against numerical results. A following experimental and numerical comparison study was performed on two interacting turbines in Mycek et al. [4]. The effect of TI on turbine thrust, power and wakes have been studied for a single turbine [5] and two interacting turbines [6]. Further experimental studies have been carried out in order to assess the flow, loads, and performance of turbines in second generation arrays [7,8]. Knowledge of the actual flow velocity distributed over a wide area in the tank is critical for such test campaigns that involve multiple tidal energy devices. In Noble et al. [8], flow measurements were performed at 150 different spatial locations, for different turbines configurations and a baseline empty tank.

While standard procedures and guidance exists for the testing of offshore renewable energy (ORE) devices and measurement of test conditions, these have been largely based on guidance for testing of ships and offshore structures [9], and they can depend on the particularities associated with the test facility [2]. In most facilities, the flow velocity varies spatially along the test tank, thus measurements at multiple locations are required to accurately characterise flow [10].

In a test tank environment, flow velocity measurements are typically performed while using non-intrusive velocimetry techniques and/or invasive sensors whose influence on the flow is minimised to avoid flow disturbance during measurement. Acoustic Doppler Velocimeters (ADV) and Laser Doppler Velocimeters (LDV) are common tools to measure mean flow velocity and turbulent parameters due to their high accuracy and high sampling frequency [11,12,13,14,15]. Both velocimetry techniques rely on the measurement of Doppler shift in the frequency of a returned signal (light or acoustic) that is back-scattered from water borne particulates. It is assumed that these particulates are moving at the same speed as the water. The Doppler shift between the transmitted and received pulse depends on the flow velocity. Multiple non co-planar beams (light or acoustic) are necessary for capturing multi-directional velocity components. Velocity measurement is performed in a small area at the intersections of the beams. Hence this can be referred as “point” measurement technique.

The disadvantage of point measurement techniques is that measurements at multiple spatial locations can only be obtained by deploying multiple devices or relocating the sensor and repeating flow conditions. In the previously mentioned tidal turbine investigations [3,4,5,6,7], flow measurements were performed with a bi-dimensional Laser Doppler Velocimeter system, capturing point measurement of the stream-wise and cross-flow velocity *u* and *v*. Flow mapping in the front and wake of the tested turbine(s) was achieved by making individual measurements on a grid using a three axis transverse system to move the light source. Other experimental investigations [8,16,17] performed flow measurements with an Acoustic Doppler Velocimeter, enabling three-dimensional (3-D) point measurements of velocity. Measurements at multiple spatial locations were achieved by mounting the ADV on an adjustable frame on the tank’s gantry (Figure 2), coupled with guy lines to reduce vibration. If the instrument is not mounted on a movable structure, the point measurement instrument requires re-positioning multiple times. Every re-positioning involves mounting and/or mooring coupled with techniques to reduce vibration and flow interference. This is particularly an issue with the ADV when measuring at difficult to access positions, e.g midway in the water column where vibration and sensor-mount interference increase. Consequently, both ADV and LDV techniques can be costly in terms of time and resource. In addition to this, a distributed measurement would provide significant advantages, such as contemporaneous measurements, used for correlation statistics and length-scale calculations, without the challenge of conducting multiple single point observations in a turbulent flow.

### Single Beam Acoustic Doppler Profiler for 1-D Multi-Point Measurements

In this study, two different types of acoustic instruments are compared: (a) Acoustic Doppler Velocimeters (ADVs) and (b) Single Beam Acoustic Doppler Profilers (SB-ADPs), introduced as a novel approach to uni-directional multi-point velocimetry in the context of tank testing. While relying on the same principles of operation as ADVs, conventional multi-beam Acoustic Doppler Profilers (ADPs) are typically deployed in order to obtain current velocity measurements in open ocean, coastal water, and rivers.

Transducers that are mounted in a diverging configuration coupled with lower carrier frequencies enable measurements that can be obtained rapidly over greater distances as compared to ADVs. Each transducer sends and receives a series of acoustic pulses. The radial velocity is estimated along the acoustic beam formed by the sound waves. The reflected acoustic signal is time gated, thereby defining a number of along-beam range cells [18]. It is assumed that the velocity within each cell (sample volume) along the profile is measured virtually simultaneously—a valid assumption given the speed of sound in water relative to the speed of the sampled current. Cells are usually regularly spread along the beam profile to facilitate processing. Radial velocity estimates are spatially averaged inside each cell using a weighting function, thus increasing sensitivity at the centre of the cell relative to the cell edge. Each cell overlaps adjacent cells, therefore adjacent cell measurements are not statistically independent [19].

Recent developments, e.g., RDI Sentinel V [20], ROWE Technologies SeaWATCH DF [21], Sontek ARGONAUT XR [22], and Signature series [19], have resulted in ADPs, which sample at higher frequencies (up to 16 Hz) and operate at higher carrier frequencies (e.g., 1 MHz), opening up the potential for these instruments to be used to increase measurement efficiency in tank facilities. In Nystrom et al. [23], two ADPs, both operating in high resolution mode (pulse to pulse coherent) were vertically mounted in a flume, pointing down the water column. Measurement of mean stream-wise velocity, Reynolds stress and turbulent kinetic energy were compared against nearby ADV measurements. The study showed good agreement between mean velocities, however this was not the case for the turbulent parameters. A Nortek Aquaddop used as a single-beam was tested in a tow tank in Harrold et al. [24]. The ADP was towed on a carriage and its ability to measure mean velocity was assessed under various carriage speeds, instrument configuration and waves used as a “disturbance”. However, no clear conclusions were drawn due to too many unknowns associated with towing the instrument.

The number of beams within the ADP designs varies from one beam [25] to eight beams [21]. Some features also enable control over the number of in-built beams used for a measurement. While the use of a single beam only enables the capture of a one dimensional velocity, it is associated with other advantages. It reduces mean insonification of unwanted regions of a tank, and avoids reflection of beams from the tank sides, floor, and the water surface, reducing interference and contamination of recorded data. This is particularly useful when measuring flow along the entire length of a tank. The diverging configuration also introduces greater uncertainty in measurements away from the instrument, as the Equations used to transform the along beam velocity to the three directional components of velocity assume that there is no variation in the flow from one beam to another [26].

A SB-ADP whose beam is precisely aligned with the desired component of the flow field can directly capture the one-dimmensional (1-D) velocity profile without the uncertainty associated with diverging beams, resulting in rapid assessment of variation in flow speed along the tank. The remaining sources of uncertainty are due to the spatial averaging within a measurement cell, depending on the beam width and the cell size [26] and the uncertainty in the estimation of the cell-average velocity, affected by Doppler broadening of the frequency shift due to random motion of the scattering particles, referred as Doppler noise. Variance of velocity measurement per ping along one beam has been estimated in Brumley et al. [27].

In order to verify the novel application of a SB-ADP to obtain measurements of 1-D velocity in a test tank, this paper presents a series of tests with an SB-ADP horizontally mounted in the University of Edinburgh’s circular wave and current test tank, the FloWave Ocean Energy Research facility [28]. The profiler was pointed into the flow to obtain the velocity component of the flow in the beam direction, i.e., incoming flow velocity. Assuming stationarity of the flow over a measurement period of two minutes [16], and repeatability of the tank flow condition, measurements from the horizontally mounted SB-ADP have been statistically compared with co-located ADV measurements performed along the profile.

Section 2 contains an overview of the experimental approach and methodology for the comparison between the SB-ADP and the ADV at different flow speeds in the tank. This includes the pre-processing and filtering of the original data as well as different statistical analysis in the time domain. The results of the analysis are presented in Section 3 and discussed in Section 4, before final conclusions are drawn in Section 5.

## 2. Materials and Methods

An inter-instrument comparison of the velocity that way measured by the ADV and SB-ADP was performed along a 8 m measurement profile, at a tank nominal velocity of 0.8 ms^−1^. This is the design velocity specification of the tank [10] and flow speed commonly used for testing 1/15−1/30 scale tidal devices. For this comparison at 0.8 ms^−1^, the ADV data were collected during the SuperGen Marine tidal array project [8,29]. In order to assess agreement across a range of speeds at which the facility typically operates, a comparison has been performed at three additional velocities from 0.6 ms^−1^ to 1.2 ms^−1^ at four measurement locations that were distributed along the profile.

### 2.1. Test Set-Up

The instruments were deployed in the circular (25 m diameter) state of the art FloWave Ocean Energy Research Facility illustrated in Figure 1. FloWave is a multi-directional wave basin incorporating current generation capabilities. The main 2 m depth testing area is separated by the rising tank floor from the lower tank area. Water is circulated via 28 drive units, each of which contains an impeller driven by a 48 kW motor and that are precisely controlled. From the impellers, the flow is driven out of guide vanes and combined to form a defined and uniform flow across the centre of the circular tank. The tank provides a 360 degree flow direction control with 168 absorbing wave makers to provide independent wave generation that can be overlaid with the current. Further detail about the facility can be found in Robinson et al. [30]. All of the tests were undertaken at nominal current velocities of between 0.6 ms^−1^ and 1.2 ms^−1^ with the wave-makers locked in the upward position, providing a water depth of 2 m. In contrast to flume tests, where target velocities for currents can be obtained for the entire length of the tank (away from boundaries), these nominal velocities represent targets for the main test area around the tank centre. These velocities are achieved via a given set of tank settings. Outside the main test area, two symmetrical eddies are forced by the impellers to stabilise the flow in the test area. Hence, the flow speed varies significantly across and along the full area of the circular tank. Tank spatial variations have been studied in detail [10,16], but they are not the matter of this paper. The focus is on the analysis of how the instruments measure this variation.

Figure 2 illustrates the test set-up. A Cartesian coordinate system is used in the tank with origin at the tank centre on the floor ($\left(x_{t},y_{t},z_{t}\right)=\left(0,0,0\right)$), $z_{t}$ vertically upwards and $x_{t}$ pointing into the main flow direction. The SB-ADP was mounted 1 m above the tank floor, on a floor bolted stand with sufficient structural stiffness to prevent vibration of the system. The structure was mounted in the dry, thanks to the 15 m diameter buoyant floor of the tank that can be lifted out of the water. The SB-ADP instrument position and orientation remained unchanged during testing with beam origin located at $\left(x_{t},y_{t},z_{t}\right)=\left(-2.98,-0.55,1.00\right)$ m (Table 1). The SB-ADP beam alignment with the horizontal plane was verified while using laser levelling and therefore the beam directly measured stream-wise velocity (*U*). The SB-ADP has a beam spread angle of 1.45°, i.e., the angle from the beam axis to the cone wall, meaning that a small misalignment of the instrument is acceptable. In Figure 2, the theoretical acoustic beam is represented as a cone, binned into measurement cells.

A movable gantry spans the tank diameter and it is typically used to suspend sensors in the flow. The ADV was mounted onto the gantry, the base of which is 1 m above the tank water surface. Gantry position can vary along the *x*-axis, and is recorded to mm precision. The measurement “point” of the ADV was aligned with the centre of the SB-ADP’s beam horizontally and vertically. To do so, the flexible head of the ADV was used, installed on a kite structure to avoid vibration of the head. This set-up enabled the ADV to capture multiple successive point measurements at multiple locations along the *x* direction profile, within the sample volumes of the SB-ADP. ADV data recorded at a nominal tank velocity of 0.8 ms^−1^ were collected during the SuperGen Marine tidal array project [8,29]. The remaining ADV data were collected specifically for the work described in this paper. The same instrument and settings were used for both measurement campaigns, only the mounting structure differed. The coordinates of the ADV measurement locations are referenced in Table 1.

Both of the instruments rely on the acoustic reflection from water-borne particulates. The tank is seeded with glass micro-beads of neutral buoyancy to ensure that there is sufficient material for backscattering of the acoustic signals. These are distributed throughout the water volume when the tank is run at speed. The ADV was removed from the water while the SB-ADP was measuring as the ADV head acts as a hard stationary scatterer that produces a stronger return than the seeding scatterers.

In this test, measurements at each velocity were made for a duration of 120 s. This sample length is greater than the tank’s flow stationarity period, defined as the period over which flow measurements have a stable mean $\bar{u}$ and variance $\sigma^{2}$, documented by [16]. To allow the tank to stabilise following a change in target velocity, a minimum of 10 min. was allowed before measurements were made at the new target velocity, as recommended in Noble et al. [10].

### 2.2. Instrumentation

The SB-ADP is a single beam variant of the Nortek Signature1000 [19] with carrier frequency of 1 MHz. It was used with a broad-band signal processing technique, where one ping is formed of two chirp modulated acoustic pulses. Broad-band processing was chosen as a good trade-off to cover both the desired spatial and velocity ranges, while enabling the resolution of short temporal and spatial scales [27]. The manufacturer states that the precision of the SB-ADP is ±0.3 cm·s^−1^ [31]. The data were recorded in burst mode, in which one single ping is recorded per sample. Burst mode enables data to be recorded at the highest sampling frequency available by the instrument (16 Hz), but recorded data are exposed to high random error [19]. Although the standard deviation of velocity measurements is inversely proportional to the cell size, it was set at the minimum possible in normal operational mode 0.2 m. These choices were motivated by maximising the amount of information collected over the short measurement range and offering the possibility for subsequent turbulence analysis from the recorded data. The blanking distance is the distance between the instrument and the first measurement cell. This is the distance (time) applied to allow the transducer, which is energised to transmit the acoustic beam, to stop vibrating (ringing) before receiving the returned echo. The blanking distance was set to the minimum available setting of 10 cm, as this can affect the first cell, but does not affect subsequent cell measurements. The relationship between the cell centre, cell size and blanking distance is expressed in Equation (Equation 1), where *n* is the cell number, $D_{b}$ the blanking distance, and $D_{c}$ the measurement cell size [19], as represented in Figure 3. The centroid, $G_{n}$ of the *n*th cell, is located at:(1)$$G_{n}=D_{b}+n\cdot D_{c}.$$

This cell covers the range $\left[D_{b}+\left(n-1\right)D_{c},\;D_{b}+\left(n+1\right)D_{c}\right]$.

The SB-ADP maximum measurement range was set as 10 m, i.e., 5.5 m before the edge of the tank. This range corresponds to the time when the instrument stops recording the return signal. The power level of the SB-ADP was set to −4 dB to reduce the energy in the transmitted pulse. This was done to limit interference from the reflected signal, and minimise acoustic reflection from the tank bottom and side, and from the water surface.

The reference instrument, the ADV, is a 10MHz Nortek Vectrino Profiler, used in non profiling mode. Table 2 lists the instrument configurations. The ADV sample frequency corresponds to the frequency at which data are recorded after averaging individual velocity estimates from multiple pings [32]. The ADV precision, as provided by the manufacturer, is ±1 mm·s^−1^ [33]. Both instruments do not require calibration, as this is performed by the manufacturer.

### 2.3. Tank Conditions

It was not possible to conduct simultaneous measurement, as the ADV head and installation frame elements act as a hard stationary scatterer for the SB-ADP. Therefore, tests were conducted at different times with the same current conditions. The repeatability of the tank is ensured by using the same tank settings i.e., motor rpm. However, this can be slightly affected by external factors, such as water temperature. Previous work has shown that the period over which flow measurements in the tank have a stable mean and variance, referred as stationarity period, was 43 s for a fixed impeller setting of ω = 82 rpm (corresponding to a target velocity of approximately 0.8 ms^−1^) [16]. The measurement period used in this paper was defined as 2 min. It was estimated to be long enough to ensure the stationarity of the flow for each nominal tank flow velocity. Hence, individual ADV and SB-ADP measurements were carried out for 2 min.

Co-located measurements were made over a horizontal profile along the x-axis, at a nominal flow velocity of 0.8 ms^−1^. This correspond to the tank impellers rotating at 96 rpm. Table 3 shows the ADV positions, herein measurement locations of the ADV collected during the SuperGen Marine tidal array project [8,29] in the *x*-direction both with reference to the tank centre ($x_{t}$) and with reference to the SB-ADP (*x*). The table also provides the number of the SB-ADP measurement cell that is co-located with each ADV measurement location and the corresponding cell centre location. The error in the *x* direction between the ADV position and the SB-ADP measurement cell centre is shown in the final column. Notice that the SB-ADP centroid of the measurement cell is given at centimetre precision, while the ADV measurement position was recorded at millimetre precision. There were 24 co-located ADV/SB-ADP measurement locations along an 8.3 m profile. SB-ADP data and additional details are contained within [34,35].

Inter-instrument comparison was conducted at three further nominal tank current velocities of 0.6 ms^−1^, 1.0 ms^−1^, and 1.2 ms^−1^, at four target locations distributed along the x-axis of the horizontal profile, in order to assess the agreement over a range of flow speeds at which the facility operates. Further details about both ADV and SB-ADP data can be found in [35]. However, following test completion and after correspondence with the instrument manufacturer, SB-ADP cell centres were found to be offset by half a cell length relative to original positioning. Accordingly, Table 4 displays the target locations A, B, C, D, i.e., the ADV measurement locations, and the corresponding pair of SB-ADP measurement cells ($n_{1}$, $n_{2}$), which overlap at the ADV measurement point location. Analysis was conducted using cell number $n_{2}$, whose centre is half a cell diameter further than the ADV measurement, cell number $n_{1}$, whose centre is half a cell diameter shorter the ADV measurement and by averaging the two adjacent cells $n_{1}$ and $n_{2}$ (herein referred to as $\bar{n_{1}\;n_{2}}$).

### 2.4. Data Processing

ADV and SB-ADP data were processed with the same suite of methods. First, a despiking algorithm [36,37] was applied while using the phase-space method based on Goring and Nikora [38]. Recommended quality control also includes correlation and signal to noise ratio thresholds. Correlation is a statistical measure of similarity in the received signal with respect to time. A common correlation threshold for the Signature is 50% [19] and 70% for the ADV [32]. Here, we applied a cut-off value of 70% for both instruments. The signal to noise ratio (SNR) is a measure of the amplitude of the signal over the background noise level. The recommendation for the ADV is to have a SNR above 15 dB [32], which was always the case. The SNR was not recorded by the SB-ADP, but it was ensured that the recorded amplitude, which measures the strength of the backscattered signal, was above 50 dB. All of the data flagged during the quality control process were replaced by scalar representations of “not a number” (NaNs), providing undefined numeric results, so as not to influence subsequent statistical analysis. Table 5 shows the percentage of data remaining after each processing step. Less than 1% of the ADV data were rejected, irrespective of the velocity at which the data were obtained. The SB-ADP correlation is more sensitive to flow speeds with 14% of data being rejected at 0.6 ms^−1^ as compared to 3% of data being rejected at 1.2 ms^−1^.

### 2.5. Analysis

For each measurement location *n* (corresponding to either a cell of the SB-ADP or its co-located ADV measurement point), the temporal mean of velocity ${\bar{u}}_{n}$ and standard deviation $\sigma_{n}$ were calculated as per Equations (Equation 2) and (Equation 3). Where $u_{n,s}$ is the instantaneous velocity of the *s*th sample at location *n*, and *S* is the total number of samples obtained over a 120 s duration, i.e., S= 12,000 measurements for the ADV and S= 1920 measurements for the SB-ADP.
(2)$${\bar{u}}_{n}=\frac{1}{S}\sum_{s=1}^{S}u_{n,s}$$
(3)$$\sigma_{n}=\sqrt{\frac{1}{S-1}\sum_{s=1}^{S}{\left|u_{n,s}-{\bar{u}}_{n}\right|}^{2}}$$

Two way analysis of variance (ANOVA) was conducted (Equation (Equation 4)) followed by a Tukey honestly significant difference (HSD) test, and a multiple comparisons of means at 95% family-wise confidence level, in order to consider the influence of more than one variable, i.e., the effect of the instrument and the sample location.
(4)$$u_{kni}=\bar{u}+I_{k}+X_{n}+I_{k}X_{n}+\epsilon_{kni}$$
where $u_{kni}$ is the measured velocity, $I_{k}$ is the instrument, $X_{n}$ is the sample location, $I_{k}X_{n}$ is the interaction between instrument and sample location, and $\epsilon_{kni}$ is the error.

## 3. Results

The results have been divided into two subsections. Section 3.1 presents the results from the inter-instrument comparison at the co-located measurements points from Table 3 for a nominal tank flow velocity of 0.8 ms^−1^. The ADV data used from the comparison at 0.8 m/s were collected during the SuperGen Marine tidal array project [8,29], the remaining ADV data and all the SB-ADP data were collected specifically for this work and are presented in detail in [35]. Twenty-seven co-located measurements were analysed along an 8 m profile, with three SB-ADP measurement cells subsequently removed after further analysis [35]. Section 3.2 presents the results from the inter-instrument comparison across a range of tank nominal flow velocities at four locations distributed across the profile. The results of the comparison for 0.6 ms^−1^, 1.0 ms^−1^ and 1.2 ms^−1^ are presented with regard to results at 0.8 m/s. By convention, the SB-ADP records negative velocity when the flow is approaching the instrument and the ADV was set with the same reference.

### 3.1. Inter-Instrument Comparison Along Measurement Profile at 0.8 m/s

Figure 4 shows for a nominal tank flow velocity of 0.8 ms^-1^, the mean and standard deviation of the co-located measurements of velocity from the ADV and SB-ADP along the profile in (**a**) and the difference between mean measurements along the profile in (**b**). The gap between 2.7 m and 3.9 m is caused by the additional analysis, as presented in [35]. The first co-located measurement indicates a higher bias than that observed on the rest of the profile (4.5% of the ADV velocity). This measurement corresponds to the first measurement cell of the SB-ADP. This region may be affected by ringing from the transducer or flow disturbance near the instrument. Hence it has been greyed. A blanking distance longer than 10 cm is recommended. Figure 4b shows overall agreement levels with regards the measurement position in the tank: absolute bias between SB-ADP and ADV is below 2.2 $\times 10^{-2}$ ms^−1^. The variation in bias along the profile requires further investigation. Possible reasons for the variation may be due to the sampling of more dynamic and complex flow regimes at the outer-most and inner-most sections of the measurement profile.

Table 6 displays the values associated with Figure 4: the mean velocity measured by the ADV ${\bar{u}}_{ADV}$, the bias in SB-ADP mean velocity measurement along the profile, and the relative bias. The relative bias is defined in Equation (Equation 5), as the difference between the temporal mean velocities measured by the SB-ADP (${\bar{u}}_{SB}$) and the ADV (${\bar{u}}_{ADV}$) as a percentage of the velocity measured by the ADV.
(5)$$Relative\;bias\;\left(\%\right)=\frac{|{\bar{u}}_{SB}-{\bar{u}}_{ADV}|}{{\bar{u}}_{ADV}}\cdot 100$$

For each co-located measurement, the mean velocity that is measured by the SB-ADP lies within the ADV standard deviation in measured velocity. The standard deviation of the SB-ADP velocity measurements is much higher than the standard deviation of the ADV velocity measurement. The standard deviation averaged over the profile is 0.1511 ms^−1^ for the SB-ADP and 0.0614 ms^−1^ for the ADV. This is due to two main reasons: (1) the ADV was sampling at 100 Hz, and the SB-ADP was sampling at 16 Hz. The total number of sample obtained by the ADV over the two minutes sampling period is much higher than the the SB-ADP, $S_{ADV}$ = 12,000 samples as compared to $S_{SB}$ = 1920 samples. (2) Within each individual sample, the ADV was internally performing averaging of velocity estimates from multiple pings $N_{p}$. Each sample collected by the SB-ADP is the velocity estimate of one ping. Based on the operation manual [19], the relationship between the standard deviation of a ping $\sigma_{i}$ and of a ping averaged sample $\sigma_{j}$ is defined as:(6)$$\sigma_{j}=\frac{\sigma_{i}}{\sqrt{N_{p}}}$$

Figure 5 displays the boxplots of the co-located measurements of velocity by the ADV (Figure 5a) and the SB-ADP (Figure 5b) along the profile. As observed in Figure 4, there is a wider spread in the velocity measured by the SB-ADP as compared to the velocity measured by the ADV. In this figure we can presume that this is due to the high level of random error that is associated with having the estimate of only one ping per sample.

The distribution of measured velocity per measurement cell as been studied in order to assess the random nature of the error in the SB-ADP ping estimated velocity and ensure that it does not introduce any bias in the mean velocity. Figure 6 shows histograms of velocity recorded by the ADV (Figure 6a) and the SB-ADP (Figure 6b) at one co-located measurement location (at x = 0.9 m). Both of the histograms show a normal distribution. This shows that the error in ping estimated velocity is random and uncorrelated from ping to ping.

Table 7 presents the results of the Two-Way ANOVA used to study the impact of two parameters on the measurement: the sample location along the profile and the type of instrument used. The *p*-values indicate that both factors are highly significant, and the hypothesis that the velocity measured is unaffected by both instrument type and sample location must be rejected. We conclude that location has a significant effect on the mean velocity, as illustrated by Figure 4. A Tukey HSD analysis was performed in order to explain the variation in space. This gives an approximate *p*-value for each measurement location. These have been plotted in Figure 7, with a colour gradient depending on the error in location between the centroid of the SB-ADP cell and the ADV measurement point. It can be seen that the error in location does not seem to influence the *p*-value. There are height co-located measurements having a *p*-value that is larger than 0.05. For those, there is no evidence to reject the assumption that both instruments are the same.

### 3.2. Inter-Instrument Comparison Across Velocity Range

Table 8 shows the bias between the velocity measured by the SB-ADP and the ADV and the relative bias as a percentage of the velocity measured by the ADV, for the target locations A, B, C and D for the nominal flow speeds of 0.6 ms^−1^, 1.0 ms^−1^ and 1.2 ms^−1^. The relative bias was calculated as in Equation (Equation 5). In addition to this, the Table highlights the three different options for calculating ${\bar{u}}_{SB}$ caused by the half cell shift centre. This shows a variation of about 1% between the three methods in calculating ${\bar{u}}_{SB}$. This Table also shows that the relative bias in velocity measured by the SB-ADP relative to the ADV varies between a maximum of 5% for a nominal flow velocity of 0.6 ms^−1^ at location A using cell $n_{1}$ to less than 0.1% for a nominal flow velocity of 1.2 ms^−1^ at locations B and D. Figure 8 displays a direct inter-comparison of the mean velocity measured by the SB-ADP and the mean velocity measured by the ADV at the measurement locations A, B, C, and D, for the nominal tank velocities of 0.6 ms^−1^, 0.8 ms^−1^, 1.0 ms^−1^, and 1.2 ms^−1^. This was produced using $\bar{n_{1}\;n_{2}}$ to calculate ${\bar{u}}_{SB}$ at 0.6 ms^−1^, 1.0 ms^−1^, and 1.2 ms^−1^, and taking four SB-ADP cells and ADV measurement locations along the profile measured at 0.8 ms^−1^ corresponding to the target locations. For each target location, the coefficient of determination $R^{2}$ is greater than 0.99. Figure 9 displays the bias in measured mean velocity between the SB-ADP and the ADV as a percentage of the velocity measured by the ADV, as a function of measurement location. Table 8 and Figure 8 and Figure 9 show that, at locations A and B, the bias in measured mean velocity by the SB-ADP largely depend on the nominal flow velocity with a large underestimation at 0.6 ms^−1^ (4.5% and 3%). For the nominal flow speeds of 0.8 ms^−1^, 1.0 ms^−1^, and 1.2 ms^−1^, the relative bias is below 2% at the four studied points.

## 4. Discussion

Using in a laboratory an instrument with profiling capabilities increases the understanding of spatial flow conditions. The SB-ADP provides a fast and simple way to measure spatial flow variation in large test facilities, by collecting simultaneous uni-directional velocity measurements at multiple locations along a profile. These series of tests have shown that, for laboratory use, the SB-ADP performs better at flow speeds that are higher than 1.0 ms^−1^, than at lower flow speeds where anomalies have been detected in post-processing. At higher flow speeds, the SB-ADP has also displayed higher returned signal amplitude and pulse correlation, both being indicators of good quality velocity data. The results have shown that the SB-ADP estimates the velocity with a difference in mean velocity varying from close to 0% to up to 2% of the velocity measured by the ADV for flow velocities of 0.8 ms^−1^ to 1.2 ms^−1^, while, at 0.6 ms^−1^, the difference varies between 0.1% and 5%. The bias introduced in the SB-ADP measurement varies with both profile range and the flow speed. In terms of range, the level of agreement decreases in the less-central regions of the tank where the velocity field is known to be most variable. Further investigation would be needed in order to understand its origins and correct for the bias, if appropriate.

A source of uncertainty in this study lies in the difference in sampling volume between the SB-ADP and the ADV. The sample volume of the SB-ADP increases with range, due to the beam spread. As the SB-ADP sample volume is much greater than the ADV, the SB-ADP may have captured some spatial variation in the flow field and then averaged it within a cell. This is likely to happen upstream of the central test area of the tank toward the edge, where the beam spread is greater and where the flow structure changes and a turbulent mixing layer occurs [30]. Despite the stiff mounting structure, SB-ADP in-built motion sensors recorded some vibration. Because of the relatively long range, small motion of the SB-ADP could have a big impact on velocity measured along the profile. This needs further investigation in order to understand the effect of vibration on measurement uncertainty.

The noise associated with SB-ADP measurement could be decreased by increasing the cell size, as there is a trade-off to be made between spatial resolution and Doppler noise. A smaller cell size leads to a shorter pulse, which induces higher Doppler noise. For a SB-ADP used in a tank to exclusively measure mean velocities without the intention to obtain turbulence parameters, the authors recommend using the SB-ADP in average mode. This will reduce the the Doppler noise by a factor of $1/\sqrt{S}$, with S being the number of samples averaged in one measurement. Standard data processing has been carried out; the authors recommend that, for the analysis of turbulent parameters, more comprehensive data processing is required to reduce the effect of the random error associated with a single ping estimate of velocity.

This sensing technique could be applied to a wide range of tank testing applications where simultaneous multi-point flow measurements over a range of the tank is required. In the presented set-up, to enable co-located measurement comparison, the SB-ADP and ADV were used individually. However, providing that there is no direct obstruction of the acoustic beam, the SB-ADP can be used in conjunction with an ADV allowing for both a multi-point profile of uni-directional velocity and a point measurement of three-directional velocity at desired location(s). Potential tank testing applications include improved wide-area tank characterisation and the measurement of bypass flow and wakes of scale model turbines. Access to multi-point data would enable more efficient tank testing and, thus, accelerate the development of marine technologies.

## 5. Conclusions

In this paper, the experimental results from a novel inter-instrument comparison between a latest-generation single beam acoustic Doppler profiler (SB-ADP) and a reference Acoustic Doppler Velocimeter (ADV) have been presented. This study has shown that a SB-ADP in broad-band mode enables multi-point measurement of current velocity along a 10 m range in a test tank, 8 m of which has been compared with co-located ADV measurements. The quality of the velocity data has been shown to be dependent on the tank flow speed and is higher at flow speeds faster or equal to 1.0 ms^−1^. In the configuration presented, the mean velocity measured by the SB-ADP data has a bias that varies between approximately 0% to 2% of the velocity recorded by the ADV for velocities faster or equal to 0.8 ms^−1^. The results are encouraging as with further optimisation of the test set-up, the results indicate the potential for this tank-range profiling technique to provide rapid acquisition of uni directional multi-point velocity measurements. Using a profiling instrument offers significant advantages over point measurement solutions, including contemporaneous measurements, and savings in terms of the time and resource associated with the relocation of a point measurement sensor and the repetition of flow conditions. Future work will investigate the sources of error to improve the accuracy of this technique.

## Figures and Tables

**Figure 1 sensors-20-03881-f001:**
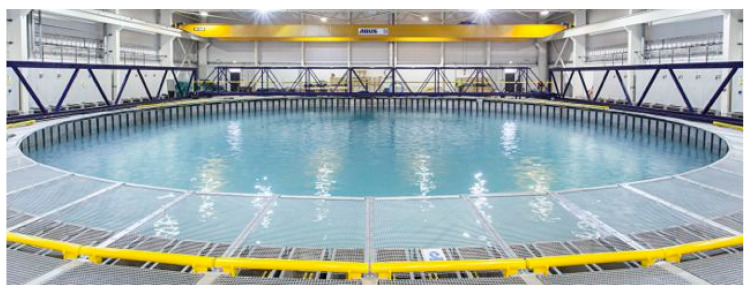
The Flowave Ocean Energy research facility - a circular wave and current test tank of 25 m diameter and a working depth of 2 m.

**Figure 2 sensors-20-03881-f002:**
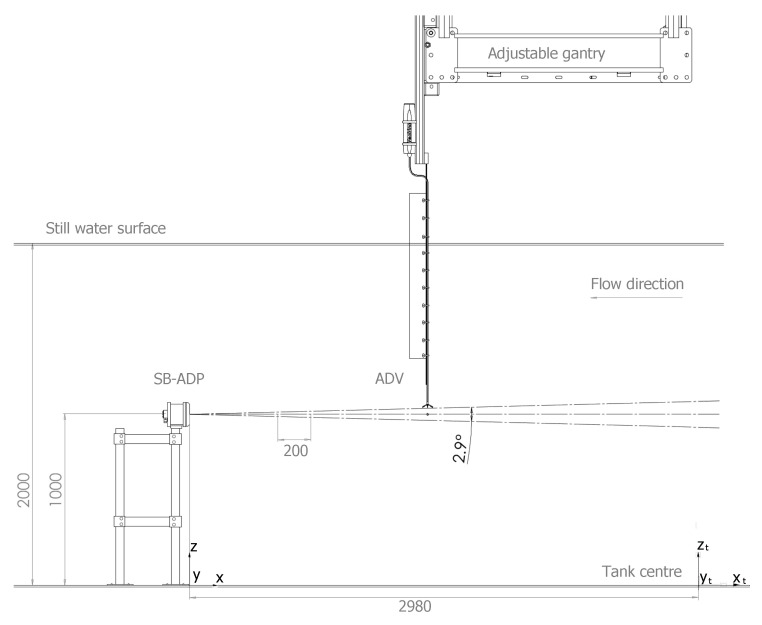
Diagram of the test set-up showing the Single Beam Acoustic Doppler (SB-ADP) pointed into the flow, and its theoretical acoustic beam binned into measurement cells. The Acoustic Doppler Velocimeters (ADV) is mounted on a x-adjustable gantry enabling point measurements within successive SB-ADP cells.

**Figure 3 sensors-20-03881-f003:**
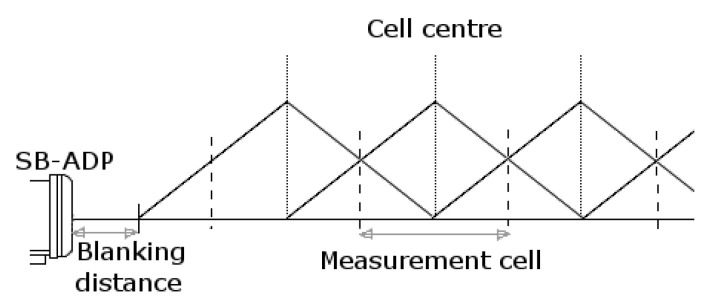
Theoretical illustration of the measurement cells of the SB-ADP.

**Figure 4 sensors-20-03881-f004:**
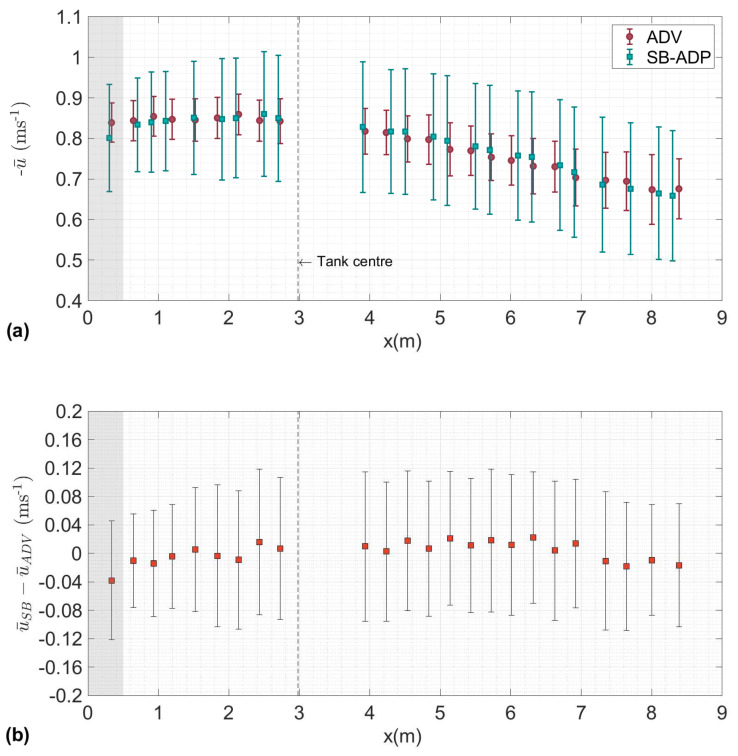
(**a**) Mean and standard deviation of the flow measured by the ADV from the SuperGen Marine tidal array project [8,29] and the SB-ADP at the cell centre for a tank nominal flow velocity of 0.8 ms^−1^ and (**b**) difference in measured mean velocities between the ADV (${\bar{u}}_{ADV}$) and SB-ADP (${\bar{u}}_{SB}$) with the difference in standard deviation. SB-ADP (${\bar{u}}_{SB}$ and ${\bar{u}}_{ADV}$) are negative due to the flow direction. The shaded area represent the region affected by the proximity to the instrument and only points with measurements conducted with both instruments are presented.

**Figure 5 sensors-20-03881-f005:**
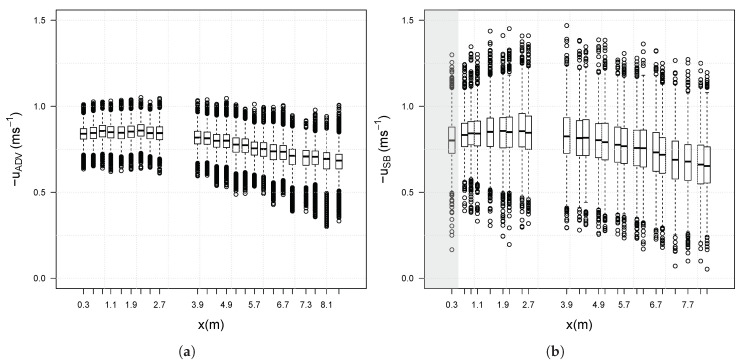
Boxplots showing the velocity minimum, maximum, median, first quartile, third quartile and outliers for the (**a**) ADV from the SuperGen Marine tidal array project [8,29] and (**b**) SB-ADP.

**Figure 6 sensors-20-03881-f006:**
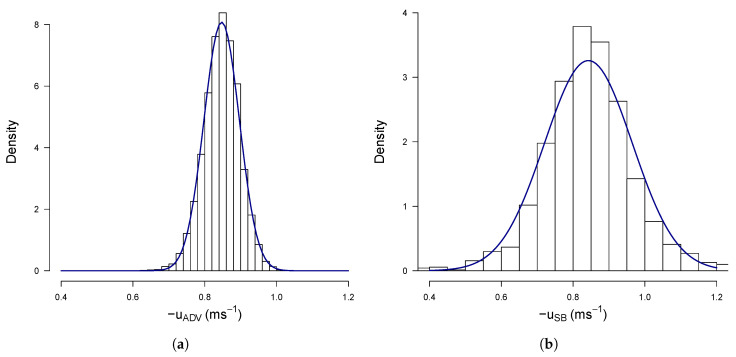
Histograms fitted with normal functions of (**a**) velocity by the ADV from the SuperGen Marine tidal array project [8,29], (**b**) velocity measured by the SB-ADP. This measurements were made at location A.

**Figure 7 sensors-20-03881-f007:**
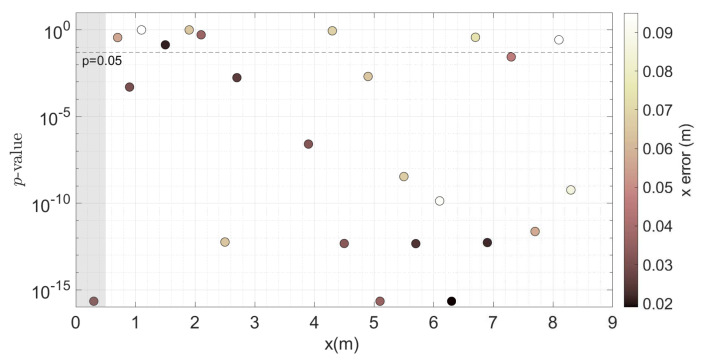
*p*-values along the profile with regard of the error in location between the centroid of the SB-ADP cell and the ADV point measurementfrom the SuperGen Marine tidal array project [8,29].

**Figure 8 sensors-20-03881-f008:**
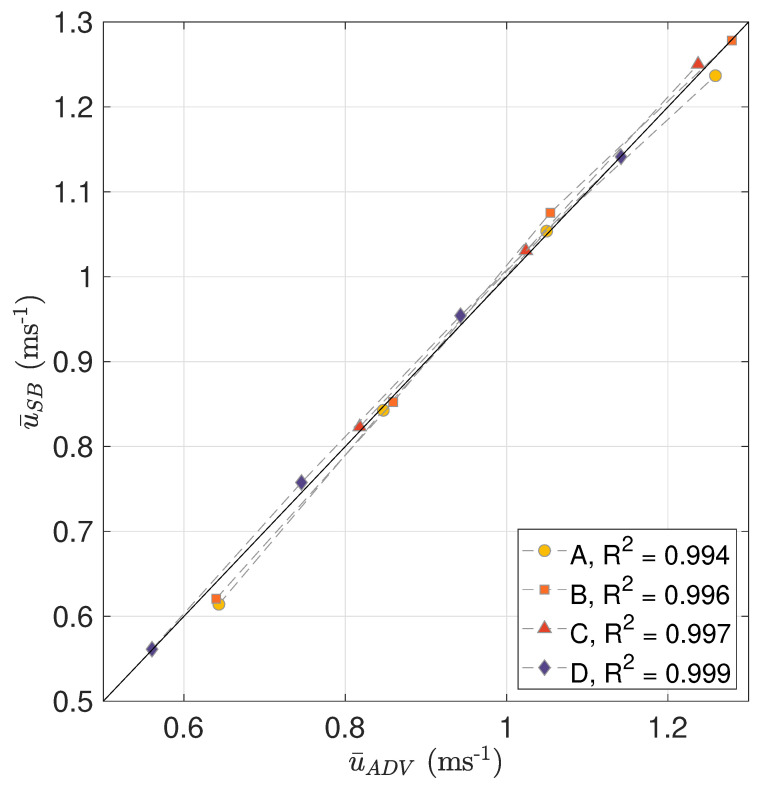
Direct inter-comparison of mean velocities measured by SB-ADP and ADV at the target locations A, B, C, and D, using the velocity of the measurement cells $\bar{n_{1}\;n_{2}}$ overlapping at the ADV measurement location. The straight line depicts a 1:1 relationship between the SB-ADP and ADV velocity measurement values.

**Figure 9 sensors-20-03881-f009:**
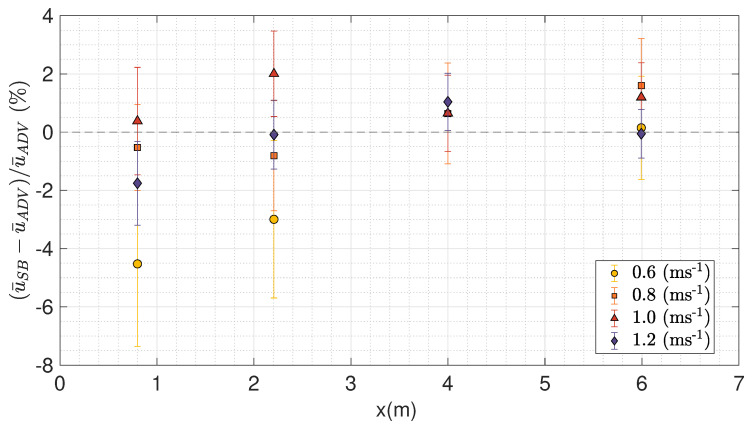
Relative bias in mean velocity measurement between the SB-ADP and the ADV as a percentage of the velocity measured by the ADV. The co-located SB-ADP and ADV measurements along the all profile were used for the tank nominal flow velocity of 0.8 ms^−1^. At the target locations A, B, C and D, for the nominal tank flow velocity of 0.6 ms^−1^, 1.0 ms^−1^ and 1.2 ms^−1^, measured velocity of the measurement cells $\bar{n_{1}\;n_{2}}$ overlapping at the ADV measurement location were used.

**Table 1 sensors-20-03881-t001:** Instruments coordinates with reference of the tank centre.

Instrument	Coordinates (m)		
	$x_{t}$	$y_{t}$	$z_{t}$
**SB-ADP**	−2.98	−0.55	1.00
**ADV**	variable	−0.55	1.00

**Table 2 sensors-20-03881-t002:** Instruments characteristics during experiments.

Instrument Type	Model	Abbreviation	Operating Frequency (MHz)	Sample Rate (Hz)	Cell size (cm)
Acoustic Doppler Profiler	Nortek Single-beam	SB-ADP	1	16	20
Velocimeter	Nortek Vectrino Profiler	ADV	10	100	0.4

**Table 3 sensors-20-03881-t003:** Measurement locations at which inter-instrument comparison was performed for a nominal tank velocity of 0.8 ms^−1^. ADV measurements locations from [8,29] are reported with reference of the tank centre ($x_{t}$) and to the SB-ADP (*x*). The SB-ADP co-located measurement cell numbers and their centre position along the *x*-axis are displayed along with the errors along the *x*-axis between the ADV measurement locations and the corresponding SB-ADP centroid.

ADV Measurement Location		Corresponding SB-ADP Cell		*x* Error (m)
$x_{t}$ (m)	*x* (m)	Number	Centroid Location *x* (m)	
−2.647	0.333	1	0.30	0.03
−2.336	0.644	3	0.70	0.06
−2.05	0.93	4	0.90	0.03
−1.785	1.195	5	1.10	0.09
−1.459	1.521	7	1.50	0.02
−1.145	1.835	9	1.90	0.06
−0.843	2.137	10	2.10	0.04
−0.545	2.435	12	2.50	0.06
−0.255	2.725	13	2.70	0.02
0.045	3.025	15	3.10	0.07
0.349	3.329	16	3.30	0.03
0.660	3.640	18	3.70	0.06
0.952	3.932	19	3.90	0.03
1.252	4.232	21	4.30	0.07
1.553	4.533	22	4.50	0.03
1.855	4.835	24	4.90	0.06
2.156	5.136	25	5.10	0.04
2.453	5.433	27	5.50	0.07
2.744	5.724	28	5.70	0.02
3.027	6.007	30	6.10	0.09
3.339	6.319	31	6.30	0.02
3.646	6.626	33	6.70	0.07
3.942	6.922	34	6.90	0.02
4.366	7.346	36	7.30	0.05
4.663	7.643	38	7.70	0.06
5.025	8.005	40	8.10	0.09
5.406	8.386	41	8.30	0.08

**Table 4 sensors-20-03881-t004:** Measurement locations at which additional inter-instrument comparisons were conducted for a range of tank velocities. ADV measurement locations are reported with reference to the tank centre ($x_{t}$) and to the SB-ADP (*x*). $n_{1}$ and $n_{2}$ indicate the SB-ADP measurement cell number immediately behind and forward of the ADV. The cell numbers are presented with their associated centre positions, as referenced to *x*.

	ADV Measurement Location		Corresponding SB-ADP Cell			
	$x_{t}$ (m)	*x* (m)	$n_{1}$	Centre *x* (m)	$n_{2}$	Centre *x* (m)
**A**	−2.181	0.799	3	0.70	4	0.90
**B**	−0.774	2.206	10	2.10	11	2.30
**C**	1.019	3.999	19	3.90	20	4.10
**D**	3.013	5.993	29	5.90	30	6.10

**Table 5 sensors-20-03881-t005:** Percentage of data remaining after data processing.

Instrument	Nominal Tank Velocity (ms^−1^)				Data Remaining after Process (%)							
					Despiking Algorithm				Correlation Threshold			
ADV	0.6		1.0	1.2	99.4		99.2	99.3	99.4		99.2	99.1
SB-ADP	0.6	0.8	1.0	1.2	98.6	98.6	99.5	99.7	86.5	90.7	96.3	97.2

**Table 6 sensors-20-03881-t006:** Mean velocity measured by the ADV along with the associated absolute bias and relative bias introduced by the SB-ADP for a nominal tank velocity of 0.8 ms^−1^. Only the SB-ADP cells that match an ADV measurement point are displayed.

**Cell Number**	1	3	4	5	7	9	10	12	13	19	21	22
**Cell Centre x (m)**	0.3	0.7	0.9	1.1	1.5	1.9	2.1	2.5	2.7	3.9	4.3	4.5
${\bar{u}}_{\mathrm{ADV}}$ **(** $\times 10^{-2}$ **ms** ^**−1**^ **)**	83.9	84.4	85.4	84.7	84.5	85.1	85.9	84.4	84.3	81.8	81.4	79.9
**Bias in** ${\bar{u}}_{\mathrm{SB}}$ **(** $\times 10^{-2}$ **ms** ^**−1**^ **)**	−3.8	−1.0	−1.4	−0.4	0.5	−0.3	−0.9	1.6	0.7	1.0	0.3	1.8
**Relative bias (%)**	4.5	1.2	1.6	0.5	0.6	0.4	1.1	1.9	0.8	1.2	0.3	2.2
**Cell Number**	24	25	27	28	30	31	33	34	36	38	40	41
**Cell Centre x (m)**	4.9	5.1	5.5	5.7	6.1	6.3	6.7	6.9	7.3	7.7	8.1	8.3
${\bar{u}}_{\mathrm{ADV}}$ **(** $\times 10^{-2}$ **ms^−1^)**	79.7	77.3	77.0	75.4	74.6	73.2	73.0	70.3	69.7	69.4	67.4	67.6
**Bias in ${\bar{u}}_{\mathrm{SB}}$** **(** $\times 10^{-2}$ **ms^−1^)**	0.7	2.1	1.1	1.8	1.2	2.2	0.4	1.4	−1.1	−1.8	−0.9	−1.7
**Relative bias (%)**	0.8	2.8	1.4	2.4	1.6	3.0	0.5	2.0	1.5	2.6	1.4	2.5

**Table 7 sensors-20-03881-t007:** 2-Way ANOVA.

	Degree of Freedom	Sums of Squares	Mean Squares	F Value	*p* Value
**Sample location**	46	1287	27.99	4571.77	$2.0\times 10^{-16}$
**Instrument**	1	0.2	0.245	39.95	$2.6\times 10^{-10}$
**Residual**	328973	2014.7	0.006		

**Table 8 sensors-20-03881-t008:** Bias in measured mean velocities between the SB-ADP and the ADV in cms^−1^, and relative bias as a percentage of the ADV measured velocity, at the target locations A, B, C, and D. The bias was calculated using $n_{1}$, the SB-ADP measurement cell having its centre ahead of the ADV measurement, $n_{2}$, whose centre is after the ADV measurement, and $\bar{n_{1}\;n_{2}}$ overlapping at the ADV measurement location.

Nominal Tank Velocity (ms^−^^1^)							
0.6 1.0 1.2							
	cell	(×10^−2^ ms^−1^)	(%)	(×10^−2^ ms^−1^)	(%)	(×10^−2^ ms^−1^)	(%)
**A**	$n_{1}$	−3.2	5.0	-	-	−2.9	2.3
	$n_{2}$	−2.6	4.0	0.4	0.4	−1.5	1.2
	$\bar{n_{1}\;n_{2}}$	−2.9	4.5	0.4	0.4	−2.2	1.8
**B**	$n_{1}$	−2.2	3.5	2.2	2.1	−0.2	0.2
	$n_{2}$	−1.6	2.5	2.0	2.0	0.0	0.0
	$\bar{n_{1}\;n_{2}}$	−1.9	3.0	2.1	2.0	−0.1	0.1
**C**	$n_{1}$	-	-	1.1	1.0	1.6	1.3
	$n_{2}$	-	-	0.3	0.3	1.0	0.8
	$\bar{n_{1}\;n_{2}}$	-	-	0.7	0.6	1.3	1.0
**D**	$n_{1}$	0.1	0.2	1.7	1.8	0.6	0.5
	$n_{2}$	0.1	0.1	0.6	0.6	−0.7	0.6
	$\bar{n_{1}\;n_{2}}$	0.1	0.1	1.1	1.2	−0.1	0.1

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
