# Peer review of "On the Use of a Single Beam Acoustic Current Profiler for Multi-Point Velocity Measurement in a Wave and Current Basin"

_sensors, 2020, doi:10.3390/s20143881_

Round 1

Reviewer 1 Report

The authors come up with an interesting contribution for flow measurement of tidal currents with the application of single beam acoustic Doppler profiler to obtain uni-directional multi-point velocimetry.

The study of renewable hydro resources, both in hydropower plants and in offshore applications, are very important in an energy transition scenario. Reference [1] is not cited within the proposal references and is a good application of acoustic Doppler profiler to measure flow for renewable generation application. Errors from several sources are analyzed, with theoretical results from CFD simulation comparison. The reference could, upon authors judgment, add value to the manuscript.

The authors used a test setup to evaluate their proposal, seeding glass micro-beads of neutral buoyancy throughout the water volume to ensure there is sufficient material for the acoustic signal reflection. In the authors judgment, what guarantee to find sufficient suspended particles in field applications?

In (5), please, avoid the use of * as symbol of multiplication because it is not a mathematical symbol. Instead, use a simple space as done in the previous equations.

Is there a scientific reason to limit the number of decimal places in the presented values? In face of the supplied data, what would be the right number of decimal figures?

Reference

Edson C. Bortoni, Luciano T. Santos, Olivier Bertrand, Patric Sauvaget, "Acoustic Doppler profiler measurements and CFD validation for Tucurui hydro power plant tailrace flow investigation," Flow Measurement and Instrumentation, Volume 68, August 2019, 101583.

Author Response

We thank the reviewer for his/her valuable work. Please see the response in the attachment. 

Reviewer 2 Report

  • The focus of this article is to analyze how the instrument measures variation. The use of relatively mature SB-ADP to measure the flow rate in the laboratory environment does not reflect the advantages of the method, it is not novel and lacks innovation.
  • “The SB-ADP has a beam spread angle of 1.45°, i.e. the angle from the beam axis to the cone wall, …”

In section 2.1, how much flow rate measurement error will result from the beam spread angle of SB-ADP.

  • The signal strength in ADV is measured by the signal-to-noise ratio SNR. If the SNR is reduced, it indicates that the noise in the ADV measurement is very large, which will affect the accuracy of the measurement data. What is the signal-to-noise ratio of this tank? Will the degree of scattering affect the measurement?
  • The signal processing method uses the existing method, there is no more detailed improvement algorithm, and the result can only show that the performance of the instrument SB-ADP is better than ADV. It is recommended to highlight the advantages of test methods and signal processing
  • It is a better choice to use boxplots method to describe the degree of dispersion of data, intuitive image.

Author Response

We thank the reviewers for their valuable work. Please see the responses in the attachment. 

Reviewer 3 Report

The article by Marilou Jourdain de Thieulloy et al. deals with the measurement of water flow velocity with a Single Beam Acoustic Doppler (SB-ADP) profiler. The authors performed a series of experiments in a test tank and compared the results with point measurements obtained using a more accurate co-located Acoustic Doppler Velocimeter (ADV). 

The structure of the article is smooth and well organized. The presented results are useful for studying similar situations and may find use in practical applications. It is a good paper in general; however, the reviewer still has some suggestions and one question listed below:

  1. In the introductory part, the task of the investigation and similar solutions are thoroughly discussed, but the conditions for conducting the experiments remain unclear for the wider readership. An additional illustration, similar to Figure 1 in [8], would improve the situation.
  2. In Instrumentation section 2.2 (lines 196 -198), it is stated that setting the measurement range 5.5 m before the edge of the tank enables the acoustic energy to fully dissipate. However, the difference of mean velocities measured with SB-ADP and ADV in Figure 3 (b) shows that it does not increase significantly over longer distances. It can be concluded that the acoustic energy is not attenuated very fast at such distances. Moreover, in [19] it is shown that the measurement range reaches hundreds of meters. Please clarify.
  3. The representation of the relative error in Table 6 is not very comprehensive because the numeric values fluctuate quite randomly. The graphical representation of the same data would probably be better traceable.

Author Response

(The authors gave the same response as above.)
